# Effects of Pharmaceutical Substances with Obesogenic Activity on Male Reproductive Health

**DOI:** 10.3390/ijms25042324

**Published:** 2024-02-15

**Authors:** Caio Mascarenhas, Ana C. A. Sousa, Luís Rato

**Affiliations:** 1School of Health, Polytechnic Institute of Guarda, 6300-035 Guarda, Portugal; caiochines@yahoo.com.br; 2Department of Biology, School of Science and Technology, University of Évora, 7006-554 Évora, Portugal; acsousa@uevora.pt; 3Comprehensive Health Research Centre (CHRC), University of Évora, 7000-671 Évora, Portugal; 4CICS-UBI—Health Sciences Research Centre, University of Beira Interior, 6201-506 Covilhã, Portugal

**Keywords:** antidiabetics, antihypertensives, endocrine disruptors, male fertility, obesogens, psychotropics

## Abstract

Obesogens have been identified as a significant factor associated with increasing obesity rates, particularly in developed countries. Substances with obesogenic traits are prevalent in consumer products, including certain pharmaceuticals. Specific classes of pharmaceuticals have been recognized for their ability to induce weight gain, often accompanied by hormonal alterations that can adversely impact male fertility. Indeed, research has supplied evidence underscoring the crucial role of obesogens and therapeutic agents in the normal functioning of the male reproductive system. Notably, sperm count and various semen parameters have been closely linked to a range of environmental and nutritional factors, including chemicals and pharmacological agents exhibiting obesogenic properties. This review aimed to explore studies focused on analyzing male fertility parameters, delving into the intricacies of sperm quality, and elucidating the direct and adverse effects that pharmacological agents may have on these aspects.

## 1. Introduction

According to the World Health organization, in 2016, more than 1.9 billion adults aged 18 and over were overweight. Of these, more than 650 million adults were obese. Obesity is considered as a chronic disease, derived from interactions between genetic factors, but also from factors associated with the lifestyle of developed societies (e.g., sedentarism, consumption of high-fat diets), and more recently, it has been associated with exposure to obesogen compounds [1,2]. These compounds specifically stimulate a rise in adipocyte (fat cell) count and disrupt metabolic and endocrine functions, including the control of appetite and satiety [3,4]. Obesogens emulate the actions of endogenous hormones, resulting in undesirable effects. These compounds attach to receptors situated in cell membranes, cytosol, or even the nucleus, thereby inducing changes in cellular responses and gene expression. Obesogens disrupt adipocytes by altering transcriptional regulators in lipid flow and/or the proliferation/differentiation of these cells, notably through the peroxisome proliferator-activated alpha receptor (PPARα), peroxisome proliferator-activated delta receptor (PPARδ), peroxisome proliferator-activated gamma receptor (PPARγ), and steroid receptors [5,6]. The main regulator of adipogenesis is PPARγ, and its activation provides differentiation of adipocytes and/or induction of lipogenic enzymes [7,8,9,10]. The adipocyte differentiation process entails a highly intricate transcriptional regulatory network, and numerous mechanisms remain unidentified, necessitating further exploration. In brief, PPARγ engages in heterodimer formation with members of the retinoid X receptor (RXR) family, binding to consensus sites marked by imperfect direct repeats of the AGGTCA sequence, separated by a single base pair (DR1 elements) [11]. The crystalline structure of the DNA-bound PPARγ-RXR heterodimer, in the presence of ligands and coactivator peptides, substantiates the functional observations that initially delineated the domains of PPARγ [12]. In the presence of a ligand, a structural alteration occurs in the ligand-binding domain of PPARγ, facilitating engagements with coactivators, including steroid receptor coactivators, histone acetyltransferases such as CBP and P300, as well as the Mediator complex. This molecular rearrangement ultimately contributes to the transcription of obesity-related genes [9,13].

Compounds like organotins (OTs) are characterized by the presence of a tin atom (Sn) covalently attached to one or more substituent groups. The organic group includes, for example, methyl, ethyl, propyl, butyl, and phenyl, and the anionic group may include a chloride, fluorine, an oxide, hydroxyl, carboxylate, or thiolate. According to the number of organic compound groups, Ots are grouped into mono-, di-, tri-, and tetrasubstituted tin compounds [14].

Among the various OTs, tributyltin (TBT) is considered the “model obesogens” as it interacts with the PPARγ [15]. TBT functions as an agonist for both PPARγ and RXR. These receptors operate as a heterodimer, robustly encouraging the differentiation and persistence of adipocytes, while also promoting the expression of genes associated with the formation and storage of lipid droplets [15]. In vivo exposure to TBT during puberty can result in weight gain, insulin resistance, elevated leptin levels, and various other metabolic effects. In adult animals, exposure to TBT leads to an increase in fat mass. However, it remains unclear whether this effect is reversible or permanent [4]. Obesogens are not restricted to organotins and encompass a list of more than 50 compounds. This list includes a diverse array of substances ranging from biocides, plasticizers, and industrial chemicals to pharmaceuticals and personal care products as well as some natural compounds including certain estrogens [16,17]. Although most studies on obesogens focus on chemicals other than pharmaceuticals, there is substantial evidence that pharmaceuticals may act as obesogens and, thus, contribute to overweight and obesity [17]. This comes as no surprise if we consider, for example, the well-established association between antidiabetic medications and weight gain. While epidemiological data indicate an association between exposure and effects, it is crucial to consider that most of these studies lack a mechanistic explanation due to their inherent limitations. Changes in weight, associated with exposure to obesogen, are a serious public health problem. Exposure to obesogen may also occur in the prenatal period, which may cause complications in the development of the endocrine system of the fetus [4]. Several epidemiological studies have indicated an association between obesogen exposure and elevated body mass index and/or the occurrence of metabolic disorders in humans [7,8,18,19]. 

Other studies have shown that certain chemicals act as PPARγ agonists. Theoretically, exposure to obesogens, even during an early age or adolescence, may lead to a permanent increase in the number of adipocytes [20]. In addition to the negative impacts of obesogens on metabolic regulation, exposure to these compounds also influences other mechanisms, including the regulation of male fertility. Typically possessing lipid characteristics, obesogens can easily traverse cell membranes. The male reproductive system is highly susceptible to the detrimental effects of these compounds as they cross the blood-testicular barrier, leading to damages to germ cells. It is important to note that, aside from the impacts of obesogens on male fertility, obesity itself can also adversely affect male reproductive health [21]. Recent reviews have highlighted the impact of obesogens on male reproductive health, e.g., [22,23,24]. However, they focused mainly on obesogenic compounds other than pharmaceuticals. This paper aims to provide an updated review, examining the influence of pharmacological agents exhibiting obesogenic characteristics on male reproductive health. The focus is particularly on the regulation of the reproductive axis, potential effects on classical reproductive clinical parameters, and the broader implications for male fertility potential.

## 2. Pharmacological Agents with Obesogenic Activity

### 2.1. Corticosteroids

Corticosteroids induce weight gain mainly through glucocorticoid (GC) therapy. About 70% of all subjects undergoing chronic GC therapy, quantified as a prednisone equivalent, significantly experience an increase in weight [25]. GC (corticosterone, see Figure 1—please see all chemical structures in the Appendix A) stimulates appetite through changes in the activity of protein kinase that is activated by adenosine monophosphate (AMP) in the hypothalamus, consequently influencing the individual’s eating habits. GC can act through the peripheral stimulation of the cannabinoid receptor 1 (CB1R) in the liver, inducing hepatic lipogenesis, steatosis, and dyslipidemia. By increasing the expression of CB1R in adipose tissue, GC induces insulin resistance (IR) and obesity.

### 2.2. Antidiabetics

Antidiabetic drugs are intended to control blood glucose levels and are, therefore, used in the treatment of diabetes mellitus (DM). Some antidiabetic pharmacological agents cause weight gain, such as the hormones insulin, sulphonylureas (SU), and some thiazolidinediones (TZD) (see Figure 1). The mechanisms of action of each compound are different and vary depending on certain conditions, such as dose and concentration. Insulin leads to a dose-dependent weight gain through multiple mechanisms [26]. Appetite stimulation, sometimes triggered by hypoglycemia or deep fluctuations in blood glucose, is probably the most important factor in increasing body weight. Thus, the anabolic characteristics of insulin can cause weight gain by inhibiting lipolysis and proteolysis [26,27]. Regarding sulphonylureas (SU) and thiazolidinediones (TZDs), weight gain seems to be more pronounced in the first months of therapy [28,29]. Additionally, SU causes hyperinsulinemia regardless of the glucose levels [30]. Similar to SU, TZDs also generate weight gain ranging from 1.5 to 4 kg in the first year of treatment [31,32]. The mechanisms through which TZDs cause weight gain include fluid retention, increased storage of lipides, and adipogenesis. 

### 2.3. Antihypertensive

Hypertension is a comorbidity associated with obesity and type 2 diabetes mellitus (T2DM). Antihypertensive therapy should be thoroughly monitored, as drugs that generate weight gain or have other metabolic effects are of concern to hypertensive individuals. β-blockers (Figure 1) used as antihypertensives display several effects on weight, they may not lead to any significant change, or they may lead to an increase of 4 kg after 1 year of treatment [33,34]. These effects consist of a decrease in the total energy expenditure of 4–9% due to β adrenergic antagonist therapy [35], since β-blockers reduce the basal metabolic rate by about 12% in obese hypertensive individuals, compared to patients receiving other antihypertensive drugs. Another effect associated with β-blockers is the inhibition of lipolysis in response to adrenergic stimulation, hindering weight loss [36]. However, some studies concluded that the use of the β-blocker carvedilol did not affect the control of glycemia and even improved some components of the relationship with metoprolol (selective medication of type β-blocker) in individuals with DM and hypertension [34,37,38]. 

### 2.4. Psychotropic

Many classes of psychotropic drugs are associated with significant weight gain and with diseases associated with obesity and its comorbidities such as diabetes, hypertension, and coronary artery disease. Psychiatric patients have a higher tendency to develop obesity when compared to the general population. Some psychiatric disorders are characterized by a chronic stimulation of the hypothalamic–pituitary–adrenal (HPA) axis and increased cortisol levels, leading to increased abdominal fat, hepatic steatosis, and insulin resistance (IR) [39,40]. The bidirectional relationship between obesity and depression is known [41]. Peripheral chronic inflammation induced by visceral fat accumulation and central nervous system inflammation are associated with pathophysiological changes in the brain related to depression [39,41]. Chronic psychosocial stress can cause inflammation and metabolic alterations, which include weight gain with a predominance of visceral fat accumulation and IR [42]. There is a positive association between weight gain and the time of exposure to psychotropic medication [43]. However, the concrete effect of drugs on weight is often difficult to quantify since psychiatric disorders can cause changes in mood, appetite, and fitness for physical activity. Many drugs of this class (see Figure 2—please see all chemical structures in the Appendix A) have the potential to interfere with central appetite-regulating neurotransmitters, altering their effects on dopaminergic, serotonergic, and histaminergic neurotransmission. Thus, through several mechanisms, these affect satiety and metabolic homeostasis [44]. Generally, the differences in the mechanisms that increase weight are partially explained by their differential affinity and effect on the different receptors of neurotransmitters, which may vary from individual to individual [43,45].

#### 2.4.1. Antidepressants

There is a large difference in weight gain during antidepressant therapy, as this varies greatly between different classes and their duration. The use of tricyclic antidepressants (TCA) is associated with weight gain both in the acute phase of the disease and in the period of maintenance of therapy [20]. For example, selective serotonin reuptake inhibitors (SSRIs) theoretically have a weight-reducing effect. This is due to their effect on serotonin, which is important in controlling carbohydrate intake. The intake of antidepressants such as amitriptyline, nortriptyline, and mirtazapine (Figure 2) is associated with a considerable weight gain [46].

#### 2.4.2. Lithium

Lithium treatment is associated with side effects associated with metabolic disorders, namely, hypothyroidism and hyperparathyroidism, thus generating weight gain. Lithium, in turn, is responsible for interfering with the production of thyroid hormones by interfering with iodine absorption. Studies have shown that 60% of patients suffering from bipolar disorder who undergo lithium therapy have a 5% higher incidence of significant weight gain. A possible mechanism triggered by lithium is direct control over the mechanisms that regulate appetite from its effects on the hypothalamus. Consequently, there is an increase in satiety by very caloric foods and beverages [47].

#### 2.4.3. Antipsychotics

Second-generation antipsychotics (SGAPs) or atypical antipsychotics (Figure 2) are often prescribed for psychotic disorders, such as bipolar disorders. These have a peculiar pharmacological profile because they induce fewer extrapyramidal effects; however, individuals who take SGAP increase their weight by at least 7% [48,49]. The effects of antipsychotics on weight gain are dose- and time-dependent, and as well as psychotropic medication, the effects on body weight can be predicted through the progression of weight variation in the first weeks of treatment. Clozapine, olanzapine, quetiapine, and risperidone are associated with a higher weight gain. Aripiprazole, amisulpride, and ziprasidone are also associated with weight gain, but it is less than the former, being classified as neutral weight agents [31,50]. 

#### 2.4.4. Antiepileptics

The most used antiepileptic drugs are valproate and carbamazepine (Figure 2), which induce weight gain in 43% of patients [51]. Pregabalin and gabapentin may also induce weight gain [52]. It has been demonstrated during valproate therapy that weight gain is higher in the first year of treatment and that women appear to be more susceptible than men [53]. The interaction of valproate with leptin and resistin, neuropeptides that regulate appetite in the hypothalamus and with effects on energy expenditure, may be the main factors responsible for weight gain [20]. Valproate can also cause other problems such as increased cholesterol and triglycerides, or decreased levels of high-density lipoprotein, HDL (dyslipidemia) [54]. On the other hand, the mechanisms of weight gain induced by carbamazepine, pregabalin, and gabapentin are not related to increased IR, metabolic syndromes, or the risk of hyperglycemia [55].

## 3. Effects of Pharmacological Agents on Male Fertility Parameters

The regulation of male fertility is a very complex process and can be interrupted by numerous factors such as life style, drug consumption, and exposure to environmental contaminants that present endocrine-disrupting and obesogenic properties [56]. In order to understand the impact of such factors, several endpoints need to be evaluated. While all animals are monitored by body weight and fertility indices, there are specifically relevant assessment points that should also be considered, including the macroscopic examination and morphology of reproductive organs, observation of development effects, and measurement of spermatozoan effects. Weight and histopathological analyses of the testicles, epididymis, and accessory sex glands, including the prostate and seminal vesicles, should be conducted because these sex glands are dependent on androgens, and these hormones reflect changes in the endocrine state of the animal or testicular function [8,56]. Additionally, as the normal physical development may also be affected by the exposure to obesogens, testicular descent, anogenital distance, and the external genitalia structure need to be evaluated. Finally, the evaluation of the number of spermatozoa, morphology, and motility is of paramount importance as the number of spermatozoa is derived from spermatid head counts in the testicle and epididymis [24,57,58,59]. Furthermore, the comprehension of the impacts of the exposure to different pharmaceutical agents is further conditioned by a latency period, as the side effects of the drugs may be observed even after the discontinuation of treatment, being necessary for 1 to 3 months for the parameters to be restored. Some studies have shown that obesogens can alter endocrine activity in various ways, affecting androgen production. The hypothalamus–pituitary–gonad axis (HPG) is fundamental for male sexual maturity and fertility regulation. The hypothalamus secretes the gonadotrophine-refining hormone (GnRH), which causes an increase in the nocturnal secretion of pulsatile gonadotrophy (luteinizing hormone (LH) and follicle-stimulating hormone (FSH)) by the anterior pituitary [56,60]. In humans, a neonatal testosterone spike during the first 4 months of life leads to testosterone levels that resemble those of a healthy adult man [61]. At 6 months of age, testosterone levels decrease to almost negligible levels and remain low until puberty, at which time sexual characteristics develop. In adult men, the pulsatile secretion of gonadotrophin occurs approximately every 90 min; the frequency with which this occurs is an important factor in the normal gonadal response [62]. When the pulse frequency of gonadotrophin reaches a critical level, secondary sexual characteristics begin to form; this marks the onset of puberty [60]. While FSH is not used until sperm maturation, LH is released during sleep along with pulsatile GnRH, causing gonadal stimulation and the induction of the hyperplasia of Leydig cells, responsible for testosterone production [63]. Any therapy that affects testicular endocrine function has as a direct consequence of the deregulation of spermatogenesis, compromising the viability of germ cells or even the integrity of Sertoli cells. Sertoli cells play a supporting role in the survival and differentiation of germ cells, as they not only provide physical support, but also ensure the “nutrition” of the germ line [45,64]. Another target of the endocrine disruption of the testicles is the production of testosterone by Leydig cells or even the disturbance of the hypothalamus–pituitary axis, which results in deficient testosterone production. Treatments with certain classes of medications (antidiabetic, antihypertensive, antidepressant, antipsychotic, and antiepileptic) can cause endocrine disorders, which end up directly interfering with the male reproductive tract and all molecular mechanisms responsible for the physiological maintenance of male fertility (see Figure 3).

### 3.1. Antidiabetics

Although certain drugs of this class have obesogenic characteristics, such as insulin, SU, and some TZDs, pioglitazone and metformin (biguanide) are the most notorious ones. These drugs are used in the treatment of T2DM, being suitable for young people and men of reproductive age. Pioglitazone is a potent synthetic agonist of PPARg-activated receptors [65]. The activation of these receptors leads to the increased transcription of genes related to glucose metabolism. Pioglitazone is available to make combinations with other antidiabetics such as metformin [66]. It differs (heterodimerizes) from retinoid receptor X and binds to responsive nuclear elements, thus modulating the transcription of genes that play a role in glucose and lipid metabolism [9]. An adverse effect that this class has is fluid retention, which makes TZDs contraindicated in individuals with heart failure, which is one of the leading causes of death in individuals with T2DM [67]. In a recent study, it was observed that pioglitazone at the dose of (10 μM) increases lactate production in in vitro cultures of human Sertoli cells without causing morphological changes or considerable metabolic changes (Table 1). This result is relevant since lactate is a survival factor of some germ cells and, thus, pioglitazone is a positive factor in testicular metabolic reprogramming [68].

### 3.2. Antihypertensive Drugs

The mechanisms by which these treatments affect sexual function are yet to be properly clarified, with limited studies providing data on the interactions between this class of drugs and sexual dysfunctions. Therefore, it is still difficult to show the impact they may have on the reproductive tract. Some studies suggest that hypertensive erectile dysfunction may be the result of reduced penile pressure, associated with decreased systemic pressure reduced by the antihypertensives themselves [69,70,71]. Other studies have shown that antihypertensives with central action stimulate α2-adrenergic presynapttic receptors, causing a reduction in the central sympathetic tone. The increase in the sympathetic tone leads to short-blooded tacardia; consequently, centrally activated antihypertensives would be a good choice for the treatment of hypertension. However, they can generate changes in ejaculatory function, as a sign of hormonal changes [69,72]. 

Other drugs, including alpha blockers and calcium channel blockers, do not cause erectile dysfunction but may generate ejaculatory disorders due to spongy bulb muscle contractions [64]. Angiotensin-converting-enzyme inhibitors and sartans, on the other hand, do not alter erectile function. In fact, angiotensin II is an important mediator of penile tumescence (erectile dysfunction). Regarding the effects of drugs that improve endothelium function in erection, it has been demonstrated that angiotensin II antagonists are very promising [73]. In fact, angiotensin II antagonists are considered a good option for the treatment of hypertension in sexually active individuals, since they improve endothelial function in penile erection [69,74,75,76,77]. Some studies on the implication of antihypertensive drugs in erectile dysfunction showed that sartans were not associated with the development of sexual dysfunction and that they might be a viable therapeutic treatment option to prevent or correct erectile dysfunctions in patients with hypertension [69,70,74,75,78,79,80]. The administration of drugs with antihypertensive action, such as calcium channel blockers, compromise fertility, particularly by reducing the viability and motility of spermatozoa (Table 1). The alteration of these clinical parameters prevents the fertilization process because it prevents the sperm and oocyte from interacting, resulting from the transmembrane movement of the calcium modifier, as described in vitro in [80,81,82,83]. Although these side effects in hypertensive individuals do not constitute a motive for therapy discontinuation, it is important to consider the beneficial effects that antihypertensives have in relation to cardiovascular problems. 

### 3.3. Psychotropics

#### 3.3.1. Antidepressants

Selective serotonin reuptake inhibitors (SSRIs) are the first-line treatment for depression, and tricyclic antidepressants play the role of inhibiting the recapture of catecholamines at the central level. Tricyclic antidepressants and SSRIs are responsible for hyperprolactinemia, which will inhibit the hypothalamus. If hypogonadism is observed, these drugs may cause adverse effects on male fertility as described below [79,80,84,85]. SSRIs are responsible for the increase in serotonin neurotransmission, thus causing an inhibition of male sexual behavior. Men who are under this type of therapy have a loss of libido and anorgasmia during prophylaxis. Delayed ejaculation, or even anejaculation, have also been observed in a few weeks after the start of treatment. Treatment with tricyclic antidepressants is associated with delayed ejaculation/anejaculation [72,75,78,79,80,86,87,88]. At the clinical level, SSRIs alter sperm count and motility (Table 1). However, there is some controversy in the literature about the adverse effects on reproductive clinical parameters. Antidepressants can also alter sperm quality through a mechanism that affects its transport, since drugs belonging to this class act at the level of the central nervous system and affect the ejaculation process [85,89,90,91]. Monoamine oxidase inhibitors (MAOIs) are also prescribed for the treatment of depression, but with a lower frequency and priority, and they rarely generate any erectile disorders, as they are less associated with hyperprolactinemia [84]. Depression is characterized as a chronic condition, and in the event of adverse effects such as changing sexual parameters, in the therapy based and MRI or MAOI, discontinuation of treatment is a very viable option. In relation to other antidepressants such as bupropion, mirtazapine, and buspirone, it was found that they do not cause considerable changes in male reproductive health [72,78,79,80,86,88,92,93,94].

#### 3.3.2. Antipsychotics

Like tricyclic antidepressants and SSIss, “typical” neuroleptics (phenothiazides) are responsible for hyperprolactinemia, the most common abnormality of the hypothalamic-hypothesis axis, caused by exaggerated production and prolactin (PRL) due to the central dopamine secretion block. As a result, one of the most relevant adverse effects are changes in spermatogenesis and the consequent decrease in sperm quality [80,84,95]. The “typical” neuroleptics can also affect sexual activity, e.g., through difficulty in reaching orgasm even with sexual stimulation (anorgasmia), decreased libido, and disorders of the reaction. Dysfunctions that are observed during treatment may also result from an anticholinergic effect induced by the antipsychotics of the β-adrenergic transmission block [64]. As psychotic conditions are often difficult to interpret, it is difficult to tell whether sexual disorders come from chronic disease or its treatment. If any sexual dysfunction appears during treatment with typical neuroleptics such as haloperidol or amissulpride, it may be possible to switch to a treatment with atypical neuroleptics such as quetiapine, olanzapine, or clozapine, as these have an affinity for 5-HT_2_ receptors and also for dopamine D_2_ receptors, facilitating sexual behavior by causing fewer side effects [72,78,80,86,96].

#### 3.3.3. Antiepileptics

Sexual dysfunction in individuals with epileptic conditions is due to several causes and may result from the pathophysiology of the disease and/or treatment with associated anticonvulsants. It is extremely difficult to distinguish the cause of these changes, as epileptic men usually rarely remain without proper treatment, which, therefore, makes it difficult to determine whether the changes are a result of epilepsy or the interaction of the drugs used in the treatment. Studies conducted in animal models have shown that valproate administration is responsible for reducing testicular weight in animals [82,97,98]. Valproate does not increase SHGH levels but increases the level of GABA in the central nervous system, thus modifying the production of GnRH. This drug may also be responsible for increased peripheral levels of androgens, and a possible reduction of LH [98,99,100,101,102]. Inducers of liver enzymes such as carbamazepine and phenytoin increase the synthesis of sex-hormone-binding globulin, the protein involved in testosterone transport, but reduce free levels of androgens [103]. On the clinical level, antiepileptics are believed to have a significant impact on spermatic parameters, and although the underlying molecular pathways are still unclear it is thought that it may be due to the interaction between pharmacological action and the molecular mechanisms underlying pathophysiology. Carbamazepine, valproate, and phenytoin reduce sperm motility by interfering with the sperm membrane, as valproate reduces the L-carnitine/T-carnitine ratio (Table 1). Similarly, carbamazepine is believed to be directly related to germ cells, inducing a greater number of necrotic germ cells in the lumen of seminiferous tubules. Thus, it is suspected that valproate is responsible for the reduction in the testicular weight in animals, indicating that the drug can be harmful not only in the level of spermatogenesis, but also in the degradation of testicular somatic cells, such as Sertoli cells [82,97,98,104,105].

**Table 1 ijms-25-02324-t001:** The effects of pharmaceuticals with obesogen action in reproductive cells, mainly in Sertoli cells and sperm.

Types of Pharmaceuticals	Samples/Subjects	Outcomes	References
Antidiabetics			
TZDs			
Pioglitazone	Human Sertoli cells	↓ lactate production	[68]
Antihypertensives			
Alpha Blockers Tamsulosin Alfuzosin	Human (Clinical trial)	Alterations in semen	[106,107]
Calcium Channel Blockers Nifedipine Amlodipine Verapamil Diltiazem	Human (Clinical trial)	Reduced fertilizing ability of spermatozoa (not under in vivo conditions)	[81,82,83]
Amlodipine	Human	↓ Sperm viability ↓ Sperm motility	[64,80,81,82,83] [81,82,83]
Antidepressants			
Tricyclic (Imipramine)	Human	↓ Sperm motility	[82,85,91]
SSRI			[82,85,91] [82,85,89] [82,97,100]
Fluoxetine	Human	↓ Sperm morphology ↓ Sperm count	
Paroxetine			
Sertraline			
Fluvoxamine			
MAOIs			
Bupropion	Human	No effects observed	[72,78,79,86,92,93,94]
Mirtazapine			
Buspirone			
Mocoblemide	Human	↓ Sperm motility under in vitro conditions	[85]
Antiephyletics			
Carbamazepine	Human	↓ Sperm motility ↓ Sperm count ↓ Sperm morphology	[64] [97,98,101,108] [82,97,100] [97,98,101,108]
Phenytoin			
Valproate			
Oxcarbazepine			
Phenobarbital			
Lithium	Human	↓ Sperm motility	[109]
		↓ Sperm count	
		↓ Sperm morphology	

Legend: MAOIs: Monoamine oxidase inhibitors; SSRI: selective serotonin reuptake inhibitors; TZDs: thiazolidinediones. Down arrows: decreased.

## 4. Conclusions

Within clinical settings where obesity and/or male fertility are considerations, it is imperative to conduct a thorough analysis of individuals’ histories. This examination should encompass potential pathologies and drug interactions. In this mini-review, data were gathered specifically on drugs possessing obesogenic properties, with a primary focus on the context of obesity pathology and an extension into the realm of male fertility. The review highlights the principal classes of drugs exhibiting obesogenic actions and their potential to induce hormonal changes that impact male reproductive parameters. Numerous studies have confirmed that the use of certain pharmacological agents leads specifically to weight gain. Concurrently, weight gain itself elicits unfavorable effects on the body. In alignment with the existing literature and the focus of this research, it is evident that both weight gain and the pharmacological agents under investigation induce various consequences for the male reproductive system. While all drugs with obesogenic activity generally have adverse effects on male fertility, certain classes stand out due to their more pronounced consequences. Notably, these include antihypertensives (such as alpha blockers and Ca^2+^ channel blockers), antidepressants (specifically serotonin reuptake inhibitors or SRIs), antipsychotics (particularly atypical neuroleptics like haloperidol and amisulpride), antiepileptics (valproate), and antidiabetics (including TZDs and SU). To advance the current research, it will be crucial in the future to address the existing gaps in information and methodological approaches used to study the impact of these agents on male fertility. Therefore, further investigations will be necessary to comprehensively understand and delineate the molecular actions of drugs on male reproductive physiology.

## Figures and Tables

**Figure 1 ijms-25-02324-f001:**
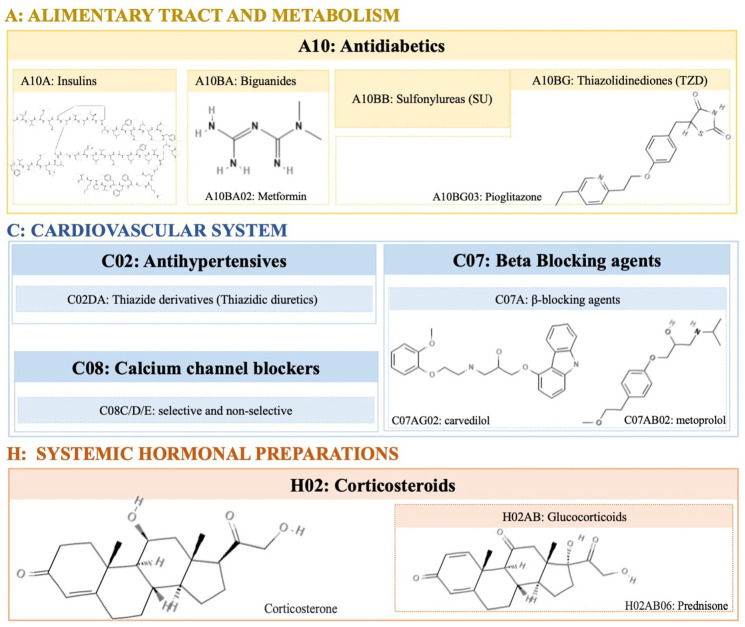
Illustration of different pharmacological agents with obesogenic activity discussed in this review, classified according to the ATC/DD Index (https://www.whocc.no/atc_ddd_index/, accessed on 28 December 2023). Molecular structures retrieved from ChemSpider and PubChem.

**Figure 2 ijms-25-02324-f002:**
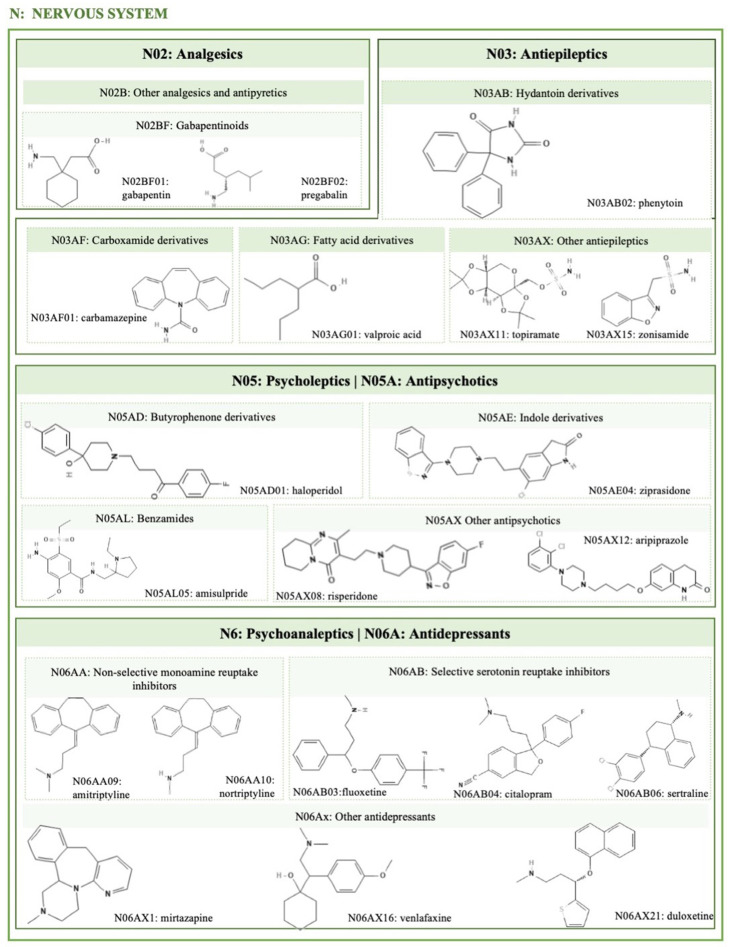
Illustration of the different psychotropic pharmacological agents with obesogenic activity discussed in this review, classified according to the ATC/DD Index (https://www.whocc.no/atc_ddd_index/, accessed on 28 December 2023). Molecular structures retrieved from ChemSpider and PubChem.

**Figure 3 ijms-25-02324-f003:**
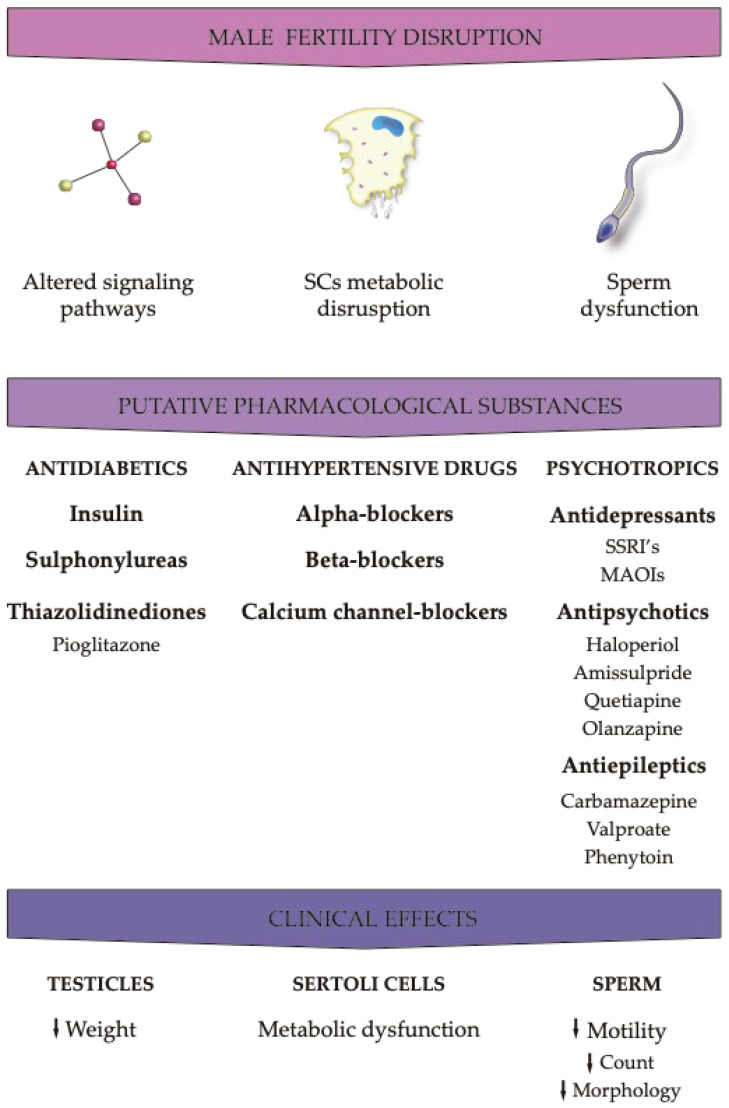
Effects of the different pharmacological agents on male fertility parameters. The effects of pharmaceuticals with obesogenic action are exerted at the peripheral level by disrupting molecular pathways that regulate male reproductive physiology. However, in addition to peripheral effects, local effects, particularly at the cellular level, have implications that can be equally or more severe than the former. Due to their relevance in the formation of germ cells, Sertoli cells are one of the main targets, as they provide physical and nutritional support to germ cells. Any action leading to the dysfunction of these cells compromises the production or the development of sperm in functionally competent cells. Down arrows: decreased.

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
