# Peer review of "Effects of Pharmaceutical Substances with Obesogenic Activity on Male Reproductive Health"

_ijms, 2024, doi:10.3390/ijms25042324_

Round 1

Reviewer 1 Report

Comments and Suggestions for Authors

In the present study, “Effects of pharmaceutical substances with obesogenic activity on male reproductive health, the authors aimed to provide an updated review, examining the influence of pharmacological agents exhibiting obesogenic characteristics on some male reproductive parameters.

As this is concerned, the manuscript has to be improved significantly as well. The manuscript illustrates and summarizes existing literature data about the pharmaceutical substances effects on reproductive axis regulation, on classical reproductive clinical parameters, and the broader implications for male fertility potential. In my opinion, the pharmaceutical agents considered are too numerous (and also belonging to different categories) to be included in a single review.

Given the great interest in the topics, this work could provide important reference information that are of good relevance, but I do not recommend the publication in its present form.

Specific comments

- Paragraph 2 is too long, as it mostly deals with the description of the individual drug. It should be reduced by at least 50%.

- Paragraph 3, on the contrary, should be expanded and accompanied by tables, reporting the reference data, and diagrams, illustrating the main effects of the various pharmacological agents on male fertility.

Comments on the Quality of English Language

Minor editing of English language required

Author Response

Ref.  Resubmission of the manuscript by Mascarenhas et al. “Effects of pharmaceutical substances with obesogenic activity on male reproductive health

Dear Editor,

Thank you for the comments to our manuscript “Effects of pharmaceutical substances with obesogenic activity on male reproductive health”. The observations made by the reviewers were very pertinent and helped us to improve the scientific outcome of the work. We have altered the manuscript as suggested by the reviewers. The alterations are marked as track changes. Below, we address each one of the reviewer comments.

Reviewer #1

In the present study, “Effects of pharmaceutical substances with obesogenic activity on male reproductive health”, the authors aimed to provide an updated review, examining the influence of pharmacological agents exhibiting obesogenic characteristics on some male reproductive parameters. As this is concerned, the manuscript has to be improved significantly as well. The manuscript illustrates and summarizes existing literature data about the pharmaceutical substances effects on reproductive axis regulation, on classical reproductive clinical parameters, and the broader implications for male fertility potential. In my opinion, the pharmaceutical agents considered are too numerous (and also belonging to different categories) to be included in a single review. Given the great interest in the topics, this work could provide important reference information that are of good relevance, but I do not recommend the publication in its present form.

Answer: The authors appreciate the reviewer’s comment and let us explain that the main idea is to show to the audience what have been done regarding the most noticeable pharmaceuticals with obesogenic effects in all sensitive points of human male reproduction, from the indirect effects (mainly how obesogens disrupt reproductive axis) to direct effects in each reproductive cell type (the most important effects and how this hamper male fertility). Please allowed us to inform, that the human studies, due to ethical issues, do not allow the same scrutiny at the molecular level, as in vivo or in vitro studies do. Furthermore, considering the type of the pharmaceuticals we are referring and their main clinical use, it is almost logical that in studies using animal models the molecular pathways governing male fertility are seldom considered of interest. In our point of view this should be, because over the last decades there has been a consistent rise in the number of couples seeking advice for fertility issues, impacting 10%−15% of the sexually active population. Neurologic conditions or any kind of these illnesses clinically diagnosed are closely associated with fertility issues, but this is often overlooked. In this regard, we brought into light the main effects of these kind of pharmaceuticals and how they affect male fertility, emphasizing its obesogenic effect. We do believe this is meritorious since some of these drugs do not hamper fertility performance directly, but via endocrine disruption mainly through obesity induction. This well documented and the correlation between obesity and male infertility is a current concern for all professionals in the reproductive area PMID: 2324876; PMID: 33044846; PMID: 34939729; PMID: 25914843. So, any physician prescribing these medicines is addressing a condition but with the detriment of other equally serious ones. Fertility complications are not determinant for life, but have a huge social impact. The association between male factor infertility and exposure to pharmaceutical compounds has been explored PMID: 23926519 but this subject sure to generate more research and discussion in the years to come. We sincerely hope the reviewer may understand our point of view and we would like to clarify the reviewer the authors added important changes in the manuscript.

Specific comments

- Paragraph 2 is too long, as it mostly deals with the description of the individual drug. It should be reduced by at least 50%.

Answer: The authors thank to the reviewer but let us explain that this in this section we aimed to present the main pharmaceuticals with obesogenic effects. The idea to only describe individually all of them is simply to put them into context and show how the substances are associated with weight gain. This section is accompanied with figure 1 and 2 in order to give graphic information about the molecules we’ve referred.

- Paragraph 3, on the contrary, should be expanded and accompanied by tables, reporting the reference data, and diagrams, illustrating the main effects of the various pharmacological agents on male fertility.

Answer: The authors thank to the reviewer and have altered accordingly. It can be found in the manuscript the new table summarizing the effects of pharmaceuticals in male reproductive cells with emphasis on sperm parameters.

We wait for your editorial decision.

With our best regards

On behalf of all the co-authors,

Luís Pedro Rato, Ph.D.

Invited Assistant Professor

luis.pedro.rato@gmail.com

Reviewer 2 Report

Comments and Suggestions for Authors

The relationship between obesity and reproductive health, especially spermatogenesis in males has been an interest for research for many years.  The authors have chosen an interesting topic and intended to summarize the current studies of advert effects of pharmaceutical substances on male reproductive health, likely via their effects on causing obesity in men.  The manuscript summarized currently known obesogens and their potential links to male reproduction.  However, there are several aspects need to be made clearer before the manuscript is acceptable for publication.

1)    Although the authors concluded that “sperm count, and various semen parameters have been closely linked to a range of …agents exhibiting obesogenic properties.”, however, little examples or studies are included in the review, it will be more convincing if the authors can summarize some experimental data and literatures that have shown the effects of obesogens on spermatogenesis and sperm functions.

2)    The figures are not mentioned throughout the main text, they should be incorporated and explained accordingly;

3)    Use abbreviations properly and explain specific terms in full when necessary, for example, lines 36-45, PPARd, PPARg, PPARg, etc.; lines 85-97, CG or GC?  Line 41, better explain what organotins are, and their relationship to tributyltin (TBT); line 128, explain what b-receptor is.  Check throughout the manuscript for the similar mistakes or improvements. 

4)    In the Introduction section, both agonist (lines 43-44) and activation effect of chemicals (lines 66-67) on PPARg were mentioned, how to understand the role of them during adipocyte proliferation?

5)    What are the evidences known that show the direct links between obesity caused by the obesogens and male reproductive health? 

Comments on the Quality of English Language

Some corrections and improvements needed.

Author Response

Ref.  Resubmission of the manuscript by Mascarenhas et al. “Effects of pharmaceutical substances with obesogenic activity on male reproductive health

Dear Editor,

Thank you for the comments to our manuscript “Effects of pharmaceutical substances with obesogenic activity on male reproductive health”. The observations made by the reviewers were very pertinent and helped us to improve the scientific outcome of the work. We have altered the manuscript as suggested by the reviewers. The alterations are marked as track changes. Below, we address each one of the reviewer comments.

Reviewer #2

The relationship between obesity and reproductive health, especially spermatogenesis in males has been an interest for research for many years.  The authors have chosen an interesting topic and intended to summarize the current studies of advert effects of pharmaceutical substances on male reproductive health, likely via their effects on causing obesity in men.  The manuscript summarized currently known obesogens and their potential links to male reproduction.  However, there are several aspects need to be made clearer before the manuscript is acceptable for publication.

Although the authors concluded that “sperm count, and various semen parameters have been closely linked to a range of …agents exhibiting obesogenic properties.”, however, little examples or studies are included in the review, it will be more convincing if the authors can summarize some experimental data and literatures that have shown the effects of obesogens on spermatogenesis and sperm functions.

Answer: The authors thank to the reviewer for this suggestion and agree with them. Please let us to explain that these changes were already addressed with the recommendations of the previous reviewer in a new table.

The figures are not mentioned throughout the main text, they should be incorporated and explained accordingly;

Answer: The authors thank to the reviewer for this suggestion and altered accordingly. Also, the figure legends were revised in the manuscript.

Use abbreviations properly and explain specific terms in full when necessary, for example, lines 36-45, PPARd, PPARg, PPARg, etc.; lines 85-97, CG or GC?  Line 41, better explain what organotins are, and their relationship to tributyltin (TBT); line 128, explain what b-receptor is.  Check throughout the manuscript for the similar mistakes or improvements. 

Answer:  The authors thank to the reviewer and have altered accordingly. New information has been highlighted in the manuscript. Regarding organotins we are glad we can clarify you by explaining that Compounds like organotins (OTs) are characterized by the presence of a tin atom (Sn) covalently attached to one or more substituent groups. The organic group includes for example e.g, methyl, ethyl, propyl, butyl, phenyl and the anionic group may include a  chloride, fluorine, oxide, hydroxyl, carboxylate, or thiolate. According to the number of organic compound groups, Ots are grouped into mono-, di-, tri-, and tetrasubstituted tin compounds. Among the various OTs, tributyltin (TBT) is considered the "model obesogens" as it interacts with the PPARγ.

. For more information we kindly suggest to see (https://www.sciencedirect.com/science/article/abs/pii/S0883292700000676) Among the various Ots, tributyltin is considered the putative agent of the best example of endocrine disruption in the wildlife, please see https://link.springer.com/article/10.1007/s10311-013-0449-8

It can be read from lines 53 to 61.

In the Introduction section, both agonist (lines 43-44) and activation effect of chemicals (lines 66-67) on PPARg were mentioned, how to understand the role of them during adipocyte proliferation?

Answer: The authors thank to the reviewer for this important observation. The process of adipocyte differentiation involves a very complex transcriptional regulatory network, and many mechanisms are still unknown and need further exploration. As referred the one comprising TBT action was defined in 2006 by Prof. Bruce Blumberg where PPRg/RXR heterodimer drives the differentiation of pre-adipocytes. Briefly, PPARγ forms an heterodimer with members of the retinoid X receptor (RXR) family, binding to consensus sites characterized by imperfect direct repeats of the sequence AGGTCA, separated by a single base pair (DR1 elements). The crystalline structure of the DNA-bound PPARγ-RXR heterodimer, in the presence of ligand and coactivator peptides, substantiates the functional observations that initially delineated the domains of PPARγ. But, even in the absence of ligand, PPARγ binds to its specific binding site. The unliganded state of PPARγ tends to facilitate interactions with NR corepressor (NCoR) and silencing mediator for retinoid and thyroid receptors (SMRT), which recruit chromatin-modifying enzymes, including histone deacetylases, to suppress transcription. On the contrary, in the presence of a ligand, a structural alteration occurs in the ligand-binding domain (LBD) of PPARγ, facilitating engagements with coactivators, including steroid receptor coactivators (SRCs), histone acetyltransferases (HATs) such as CBP and P300, as well as the Mediator complex. This molecular rearrangement ultimately contributes to the transcription of obesity-related genes. We would like to thank you again for this pertinent commentary. It can be read from lines 41 to 52.

What are the evidences known that show the direct links between obesity caused by the obesogens and male reproductive health? 

Answer: The authors thank to the reviewer’s commentary and allowed us explain that all substances, alone or combined, classified as obesogenic are intimately associated with weight gain. It is a fact that most of the signaling disruption must be elucidated, but Casals-Casas and collaborators (PMID: 21054169) have shedding light on these allowing us to infer about it in some cases and/or conditions. Besides any metabolic disturbances, these substances may exert its effects in a direct manner at all levels on the male reproductive axis. We have contributed with comprehensive reviews in the last few years showing, where and how obesogens adversely affect male fertility, please see PMID: 35328463 ; PMID: 34940512 ; PMID: 27776203 . In brief, most of these obesogens mimic endogenous hormones, competing for the hormone receptors, thus hijacking the endocrine systems (local or peripheral) Please see: PMID: 26544531. In one way or another human systems, specially those more susceptible to the toxic effects of drugs becomes dysfunctional. the testicular physiology and metabolism that are pivotal for spermatogenesis. The disruption of these tightly regulated metabolic pathways leads to adverse reproductive outcomes. Compelling evidence suggest that toxic effects of obesogens are in part mediated through metabolic disruption between testicular cells, with deleterious consequences for germ line PMID: 26544531. It is hypothesized that toxic-disrupting effects of obesogens affect testicular cells metabolic cooperation please see PMID: 26544531; PMID: 28993852.

We hope to have provided you all the needed information regarding the work submitted and that the manuscript is now suitable for publication.

We wait for your editorial decision.

With our best regards

On behalf of all the co-authors,

Luís Pedro Rato, Ph.D.

Invited Assistant Professor

luis.pedro.rato@gmail.com

Round 2

Reviewer 1 Report

Comments and Suggestions for Authors

The authors only partially satisfied my requests. Paragraph 2 is still too long. It should be considerably reduced

Comments on the Quality of English Language

Minor editing of English language required

Author Response

Dear Reviewer,  

have altered the manuscript as suggested by the reviewers. The alterations are marked as track changes. Below, we address each one of the reviewer comments.

The authors only partially satisfied my requests. Paragraph 2 is still too long. It should be considerably reduced

Answer: The authors appreciate the reviewer’s comment and have reviewed according to the suggestions. The section regarding the pharmaceuticals and weight gain was reduced and we hope is now according as reviewer’s suggestion. Please find between lines 99 to 242.

Reviewer 2 Report

Comments and Suggestions for Authors

The authors have made improvements for the manuscript.  A couple of points should be attended before publication:

1)    Refine Table 1, such as: Samples/Subjects column, not clear it’s referring to types of spermatogenic cells or infertility diseases, make it clear it’s human SC (Sertoli cells?), or spermatogonia, spermatozoa, etc.  Also clarify “viability, motility, morphology, count” as “sperm viability, motility, morphology, count”.  Mention Table 1 in the main text. 

2)    Some language editing may be needed.

Comments on the Quality of English Language

Some editing upon acceptance.

Author Response

Dear Reviewer,

We have altered the manuscript as suggested by the reviewers. The alterations are marked as track changes. Below, we address each one of the reviewer comments.

The authors have made improvements for the manuscript.  A couple of points should be attended before publication:

1)    Refine Table 1, such as: Samples/Subjects column, not clear it’s referring to types of spermatogenic cells or infertility diseases, make it clear it’s human SC (Sertoli cells?), or spermatogonia, spermatozoa, etc.  Also clarify “viability, motility, morphology, count” as “sperm viability, motility, morphology, count”.  Mention Table 1 in the main text. 

Answer: The authors thank to the reviewer and have altered accordingly. The new version of the table can be found in the manuscript. The major alterations regarding the most common clinical sperm parameters were corrected to: sperm viability, sperm motility, sperm morphology and sperm count.

The authors would also like to refer that all manuscript was entirely revised to correct minor errors.

Round 3

Reviewer 1 Report

Comments and Suggestions for Authors

The authors satisfied my requests. The manuscript, after this last revision, can be accepted for publication